# Evaluation of Disparities in Adults’ Macronutrient Intake Status: Results from the China Health and Nutrition 2011 Survey

**DOI:** 10.3390/nu13093044

**Published:** 2021-08-30

**Authors:** Yajie Zhao, Tetsuya Araki

**Affiliations:** International Agro-Informatics Laboratory, Department of Agricultural Sciences, Graduate School of Agricultural and Life Sciences, University of Tokyo, Tokyo 113-8657, Japan; aaraki@g.ecc.u-tokyo.ac.jp

**Keywords:** macronutrient composition, Dietary Reference Intakes (DRI), northern and southern regions, socioeconomic status, noncommunicable diseases

## Abstract

Little is known about the macronutrient intake status of adult Chinese people. This cross-sectional study assessed the macronutrient intake status of adults (aged ≥20 years) by comparing their intake level of macronutrients against the Dietary Reference Intakes (DRI). It further explored the associations between macronutrient intake status and age groups, genders, education levels, smoking status, drinking frequency, social classes, knowledge of Chinese Dietary Guidelines 2016 (CDGs), healthy diet priorities, and areas (urban and rural) within two regions (northern and southern). The analysis includes the dietary intake data of 7860 Chinese adults, with complete data entries in the China Health and Nutrition 2011 survey. Dietary data were obtained through the 24 h recall method. More than half had carbohydrate intake below the recommended level of intake, and more than half had fat intake above the recommended level of intake. There were significant associations between three macronutrient intakes and education levels, social classes, healthy diet priorities, areas, and regions. Disparities in macronutrient consumptions revealed geographical and socioeconomic variations in dietary patterns, as well as risks for many different noncommunicable diseases. Public health and nutrition interventions should take notice of regional differences in dietary intake and place more emphasis on vulnerable populations including women, rural residents, and people with lower education level.

## 1. Introduction

Macronutrients play an important role in sustaining lives and regulating overall human health [1]. The associations between the prevention of noncommunicable diseases (NCDs) and the proportions of these macronutrient intakes have been well documented [2,3,4,5,6,7,8,9,10]. For instance, evidence shows that diets low in saturated fat can improve cardiovascular health compared to the ones high in saturated fat [7]. In particular, one study has suggested that compared to the 55–60% carbohydrate diet, a lower-carbohydrate diet has positive effects on weight control and reduces the risk of Type 2 diabetes [5]. Similarly, the Mediterranean diet with an MUFA: SFA > 1.6 ratio of dietary fat intake can also reduce the risk of Type 2 diabetes [7,8]. In addition, many studies have shown that diets consisting of high amounts of low-quality carbohydrates could be associated with high glycemic index and mortality caused by all kinds of NCDs [9,10]. Over the past decades, diets high in carbohydrate proportion have been replaced by the ones with higher fat proportions around the globe [11]. The changes in macronutrient composition are also linked to the burden of increase in NCDs such as diabetes and overweight [11,12].

In China, it is well known that different cultural and dietary habits have been formed between the northern and southern regions of China over a long period of history [13]. Geographically speaking, the Yellow River cultivated a fertile land that gave birth to the north civilization and diet, and countless river basins and branches from the Yangtze River altogether nurtured the profound food cultures in the south [14]. Carbohydrates used to dominate the traditional northern diet in the form of wheat, as compared to rice in the southern diet [13,14,15]. Along with its economic growth and urbanization since the beginning of the 21st century, the nutrition epidemiological transition in China has been taking place rapidly across both regions [13,16,17,18,19]. Although the association between dietary intake and NCDs is well established by numerous studies based on the China Health and Nutrition Survey [20], little is known about the macronutrient intakes and the factors influencing the intake of macronutrients among adults in two geographic regions in China. Thus, it is important to understand the overall dietary intake and nutritional status of Chinese adults through examining the macronutrient intakes by regions. It is also important to have a contextual understanding of the regional differences in intakes by various sociodemographic, socioeconomic, lifestyle, and health factors. More detailed evidence may further facilitate Chinese health policymaking by targeting different population groups among different regions.

The present study undertook the call to investigate whether the macronutrient intakes among Chinese adults meet the recommended level of Dietary Reference Intakes (DRI) included in the Chinese Dietary Guidelines 2016 (CDGs 2016) [21]. It also aimed to evaluate the association between macronutrient intakes of adults (aged ≥20 years) and different age groups, genders, regions, areas, education levels, social class, smoking and drinking status, knowledge status of CDGs, and priorities in healthy diets.

## 2. Materials and Methods

### 2.1. Study Design and Sample Collection

The China Health and Nutrition Survey (CHNS) is an ongoing survey that covers myriad demographic, socioeconomic, and public health risk factors at both individual and household levels across fifteen different provinces and autonomous cities [17]. It is an international project coordinated by the Carolina Population Center at the University of Carolina at Chapel Hill (UNC-CH), the National Institute for Nutrition and Health (NINH) at the Chinese Center for Disease Control and Prevention (CCDC). Approval for using the CHNS survey data was granted by the University of Tokyo (approval number: 21-71).

The sample data were collected from nine provinces in a multistage, random cluster-drawing process in 1989. Later, nine additional data collection waves were conducted in 1991, 1993, 1997 (Heilongjiang province was added), 2000, 2004, 2006, 2009, 2011 (three megacities, Beijing, Shanghai, and Chongqing, were added), and 2015 (Shaanxi, Yunnan, and Zhejiang were added). Note that the CHNS dietary data were collected in 1991 and subsequent survey waves up to 2011 [17]. This present study used dietary data collected in 2011.

Data quality checks were performed to eliminate outliers (i.e., out-of-range values that are above the 75th percentile and below the 25th percentile of the interquartile range multiplied by a factor of 1.5). The following description shows the details of nutrition survey data collection procedures.

### 2.2. Dietary Data Collection

The CHNS survey used the 24 h recall method to collect nutrition and dietary data on a basis of three consecutive days at both individual and household levels. Individual food composition was determined by a weighing technique that measured changes in food inventory from the beginning to the end of each day. All remaining foods at the end of the day were weighed and recorded. Preparation waste (spoiled foods and discarded meals fed to pets or animals) was estimated when weighing was not possible [18]). Individual dietary data were collected for three consecutive days. Trained field interviewers used food models and pictures during the interview, asking each participant to report the amounts and types of all foods consumed at home or away from home during the previous day [18].

Despite a few cases where some one-day dietary reports were missed, about 99% of overall sample data was completed for three consecutive days. Moreover, the initial data cleaning was performed in the CHNS survey which deleted duplicated, missing, and unrealistically reported values, although some extreme values remaining in the data are left to the discretion of researchers [18].

### 2.3. Assessment of Macronutrient Intake

Based on the Food Consumption Table, the CHNS obtained the three-day average values for total daily energy intake (kcal), carbohydrates (g), fats (g), and proteins (g). The total energy intake is the sum of the three macronutrient intakes multiplied by an energy conversion factor of each: 1 g carbohydrate = 4 kcal, 1 g fat = 9 kcal, and 1 g protein = 4 kcal [22].

The Chinese Dietary Reference Intakes (DRI) is the dietary target for the Chinese population to achieve an adequate intake of nutrients. It is released by the Chinese Nutrition Society (CNS) and is constantly updated with the latest WHO recommendations and nutritional evidence. [23]. According to the DRI, the recommended daily intake of carbohydrates should account for 55–65% of daily energy intake; the recommended amount of fat intake should be 20–30% of total energy intake; the adequate protein intake should account for 11–15% of total energy intake. These cut-points are helpful to assess whether the macronutrient intake levels of each Chinese adult meet the standard intake levels.

### 2.4. Other Variables

For the purpose of this study, age was stratified into three age groups (younger adults: 20–39 years, middle-aged adults: 40–59 years, and older adults: >60 years). Gender was divided into male and female. This present analysis selected eight provinces that can be evenly divided into two geographic regions (north: Liaoning, Heilongjiang, Shandong, and Henan; south: Jiangsu, Hubei, Hunan, and Guangxi) [13]. Since some evidence has demonstrated that urban development could have an impact on the food and nutrition intakes as well as an overall health outcome [21,23], this present study further analyzes the macronutrient intake between urban and rural areas with these two regions as well.

Education level was reported as the highest degree achieved and was recoded into three different categories: low level (illiteracy; below high school), middle level (high school diploma; technical degree), and high level (college degree and above). Since the frequency of alcohol drinking was previously documented to be correlated with dietary intake [24], self-reported alcohol intake frequency was represented by three different levels: low frequency (no more than twice a month), middle frequency (1–4 times per week), and high frequency (almost every day). One dummy variable was used to represent current smoking status (no = 0, yes = 1). Social class was classified into two classes, defined by the types of occupation: high social class (i.e., nonmanual, professional, and managerial occupations), and low social class (i.e., skilled and unskilled manual occupations) [25]. Participants in the CHNS study were also asked if they know about CDGs (no = 0, yes = 1). This analysis included the participants’ knowledge of CDGs to testify if there is any relationship between the status of knowing CDGs and the compliance with DRI. Additionally, the CHNS also included a five-point Likert scale of self-rated healthy diet priority. In order to assess whether an individual’s healthy diet priority can influence his or her macronutrient intake [26], this study also included one variable that summarized participants’ perception of a healthy diet as not important, important, and very important.

### 2.5. Statistical Analysis

RStudio version 4.0.3 (R Foundation for Statistical Computing) was used to conduct all the data analysis in this study. The 3-day average total energy intake (kcal), macronutrient intake (g), and percentage of each macronutrient intake (%) were reported as means and standard deviations to represent the values of normally distributed data. Margin plots were adopted to show the relative macronutrient intakes for urban and rural areas between the south and the north. The chi-square test was used to examine the association between the DRI levels (below, meeting, and above) for macronutrients and all the predictors. A correlogram was created to display the correlation between variables of relative macronutrient intakes. One-way ANOVA was used to explore whether the associations between relative macronutrient intake and all the predictor variables were statistically significant. Post hoc tests (using the Holm correction to adjust p) were performed after ANOVA to observe which groups were significantly different from one another. Univariate simple linear regression was constructed to explore the specific correlation between relative macronutrient intakes and all the predictor variables. Three multiple linear regression models were created for three macronutrients to further analyze these associations: the carbohydrate model adjusted for age, gender, current smoking status, and CDG knowledge; the fat model adjusted for CDG knowledge; the protein model adjusted for gender, current smoking status, and CDG knowledge. Statistical significance was determined as *p* value < 0.05.

## 3. Results

### 3.1. Participants’ Characteristics

There were 7860 Chinese adults aged ≥20 years recruited in the 2011 CHNS who had complete and valid dietary data. Of these participants, 22% (n = 1686) were aged 20–30 years, 47% (n = 3720) were aged 40–59 years, and 31% (n = 2454) were aged ≥60 years. The total number of participants from northern and southern regions was 3821 (49%) and 4039 (51%), respectively. Table 1 summarized the total 3-day average energy intake (kcal), three macronutrient intakes (g), and energy from three macronutrients (%) classified by age group, gender, region, area, education level, current smoking status, drinking frequency, social class, CDG knowledge, and healthy diet priority.

### 3.2. Total Energy, Carbohydrates, Fat, and Protein Intakes

In general, according to Table 1, older adults aged ≥60 years were having the least amount of total energy and three macronutrient intakes compared to young and middle-aged adults. Males had higher intakes in total energy and three macronutrients than their female counterparts. Compared to those living in the south, people from the north had lower intakes in total energy and three macronutrients. Current smokers had higher total energy and three macronutrient intakes than those who were nonsmokers. Adults who consumed alcohol at a high frequency had higher total energy and three macronutrient intakes than those who drank less frequently. Compared with those who did not know about CDGs, participants who had knowledge of CDGs had higher intakes in total energy and three macronutrients. Adults to whom the priority of a healthy diet was high had the most total energy and three macronutrient intakes.

As illustrated in Table 1, rural adults consumed lower amounts of total energy and carbohydrate than their urban counterparts. People with low and middle education levels had higher total energy intake than those with a higher education level. Carbohydrate intake among those with a low education level was the highest. The total energy and carbohydrate intakes among adults from the lower social class were higher than among those from the higher social class.

Compared with the consistent results regarding total and carbohydrate intakes, an inconsistency was detected in fat and protein intakes. People living in rural areas had higher fat intake but lower protein intake than those living in urban areas. Participants with a low educational level had lowest fat and protein intakes than those with middle and high levels of education. Adults with a low social class had lower fat intake but higher protein intake than the ones with a high social class (Table 1).

### 3.3. Relative Energy Intake from Carbohydrates, Fat, and Protein (%)

The mean relative intake of carbohydrates was slightly lower among adults who were younger, male, currently smoking, and had high priority of healthy diet. No differences were observed in the mean relative carbohydrate intake between those who had knowledge of CDGs and those who did not. People with a high education level had the lowest relative intake of carbohydrates. Although relative carbohydrate intake did not differ between the groups with low and middle frequencies of alcohol consumption, those with a high drinking frequency were found to have a lower mean intake of carbohydrate. Compared with the mean relative carbohydrate intake, an opposite trend of fat intake was observed among adults with different levels of education, social classes, and priorities of a healthy diet. No major differences were observed in the mean relative protein intakes among adults concerning all different categories (Table 1).

### 3.4. Percentage of Adults Meeting the DRI for Carbohydrates, Fat, and Protein (%)

The percentage of adults in the study sample with relative macronutrient intakes below, meeting, and above the DRI, classified by age groups, gender, regions, areas, education levels, current smoking status, drinking frequency, social classes, knowledge of CDC, and healthy diet priority, is shown in Figure 1.

More than half of the adults had relative carbohydrate intakes below the recommended level of DRI, while more than half of them had relative fat intakes above the DRI level. About one-third of the adults had relative carbohydrate and fat intakes meeting the DRI levels, and more than half of the adults’ relative protein intake met the DRI.

There were no significant differences in the percentage of adults with macronutrient intake levels below, meeting, and above the DRI with respect to different genders (carbohydrate *p* = 0.7, fat *p* = 0.02, protein *p* = 0.46) and current smoking status (carbohydrate *p* = 0.75, fat *p* = 0.15, protein *p* = 0.96). However, there were significant differences in the percentage of adults with macronutrient intake levels below, meeting, and above the DRI with respect to different regions, areas, education levels, social classes, and healthy diet priorities (*p* < 0.001).

Approximately 14% out of the 7860 adults simultaneously met the DRI for all three macronutrients. Table 2 shows that none of the younger adults simultaneously met the DRI for all three macronutrients. The chi-square test indicated that were no significant differences among regions (*p* = 0.69) and social class (*p* = 0.28) in terms of simultaneously meeting the DRI for three macronutrients. However, there were significant differences in the percentage of adults with all three macronutrient intake levels meeting DRI with respect to different age groups, areas, education levels, smoking status, drinking frequencies, CDG knowledge, and healthy diet priorities (*p* < 0.001).

### 3.5. Consumption Correlations among Relative Macronutrient Intakes

Figure 2 is a correlogram displaying the correlations among total daily energy intake, three macronutrient intakes, and three relative macronutrient intakes. There was a strong positive correlation between daily total energy intake and carbohydrate intake, suggesting that carbohydrate was the main source of energy in daily diets. A strong negative correlation was found between relative carbohydrate intake and relative fat intake. No significant correlations were found between total energy intake and protein intake, as well as between relative carbohydrate intake and relative protein intake.

### 3.6. Linear Regression Models

Significant differences were found to exist in relative macronutrient intakes among different ages, genders, regions, areas, and levels of education, smoking status, drinking frequency, social class, and health diet priority. The specific associations between the relative macronutrient intakes and all the predictor variables were further explored using linear regression models (Table 3).

The univariate regression analysis suggested that age, gender, smoking status, and CDG knowledge (*p* > 0.05) had no effect on the relative intake of carbohydrates. The adjusted multiple linear regression revealed significant associations between carbohydrate intake and regions (*p* < 0.001, 95% CI −3.67, −2.67), areas (*p* < 0.001, 95% CI 5.23, 6.25), education levels (*p* < 0.001), drinking frequencies (*p* < 0.001), social classes (*p* < 0.001, 95% CI 3.03, 4.06), and healthy diet priorities (*p* < 0.005). Compared with adults who had low education level, those with middle and high education levels had lower intakes of carbohydrates (2.90% and 5.49% lower, respectively).

Adults living in the south had 3.17% lower carbohydrate intake than those from the north. Higher relative carbohydrate intake was associated with lower social class (3.55% higher). Adults with low and middle drinking frequencies had higher carbohydrate intake than those with high drinking frequency (3.69% and 3.40% higher, respectively). Compared to the adults living in the urban area, those from the rural area had 5.74% higher carbohydrate intake than those living in the urban area. There was an inverse association between relative carbohydrate intake and priority levels placed in a healthy diet, in which higher carbohydrate intake was associated with lower priorities placed in a healthy diet.

No significant difference was found in relative fat intake between adults who had knowledge of CDG and those who did not (*p* = 0.57, 95% CI −0.74, 0.41). However, strong associations with relative fat intake were found among age groups and smoking status (*p* < 0.005). Stronger associations were found in genders, regions, areas, education levels, social classes, and healthy diet priorities (*p* < 0.001). Interestingly, higher relative fat intake was associated with females (1.11% higher), low drinking frequency (1.04% higher), and high social class (3.00% higher). Compared to those with low education level, adults with middle and high education levels had higher relative fat intake (2.00% and 4.04% higher, respectively). Adults living in the southern region had 2.86% higher fat intake than those living in the northern region, and rural residents had 4.41% lower fat intake than their urban counterparts. Compared to adults with low priority of healthy diet, those with middle and high priority had higher relative fat intake (1.69% and 1.60% higher, respectively).

As far as relative protein intake was concerned, there were no statistically significant differences examined in different genders, status of smoking, and status of CDG knowledge. However, strong associations were observed between relative protein intake and age groups, regions, areas, education levels, drinking frequencies, social classes, and healthy diet priorities (*p* < 0.001). Compared with adults above 60 years of age, middle-aged and younger adults had higher relative protein intake (0.32% and 0.72% higher, respectively). Adults living in the southern region had 0.37% higher protein intake than their northern counterparts. Rural adults had 1.50% lower protein intake than urban adults. Relative protein intake was also associated with higher education levels, higher drinking frequency, higher social class, and higher healthy diet priority.

## 4. Discussion

This cross-sectional study examined the association between the status of three macronutrient intakes and age groups, genders, regions (northern and southern), areas (urban and rural), education levels, smoking status, drinking frequencies, social class, knowledge of CDGs, and priorities of healthy diet among Chinese adults aged above 20 years. Our results showed that more than half of the adults met the recommended level for protein intake, but the status of carbohydrate and fat intakes is concerning. Another interesting finding regarding the dietary pattern among Chinese adults was the inverse relationship between the relative intake of carbohydrates and that of fat. In line with the same finding, one study concluded that high carbohydrate intake may be associated with lower fat intake [27].

Moreover, there were significant differences in the relative intakes of the three macronutrients with respect to different regions, areas, education levels, social classes, and healthy diet priorities. Some very interesting findings were provided by this current analysis. First, more than half of the adults had carbohydrates intake below the recommended level, with adults in the high education level and social class having less carbohydrate intake than those in the low education level and social class. Meanwhile, more than half of the adults had fat intakes above the recommended level, with adults in the high education level and social class having more fat intake than those in the low education level and social class. These findings were compatible with results of previous studies suggesting that the dietary pattern of those with better socioeconomic status has shifted to the high-fat and low-carb macronutrient proportion, which could be one of the major causes of the nutritional epidemics in China [28]. Furthermore, a similar study conducted in Norway also reported the link between education gradients and macronutrient intake, in which higher education was associated with higher fat and lower carbohydrate consumption [29]. Therefore, the diet and health interventions should focus on improving education level with an emphasis on less-advantaged groups (i.e., those with lower socioeconomic status and education level).

Second, lower carbohydrate intake and higher fat intake were also found in urban adults living in the southern region compared to their northern counterparts. This finding was consistent with previous literature suggesting that economic growth, urbanization, and globalization of food can play an important role in affecting eating habits and prevalence of NCDs [30,31,32]. Despite the fact that southerners had higher relative fat intake, studies have shown that hypertension and obesity were more common in the northern region [32]. This could be explained by the higher MUFA:SFA ratio in the southern diet that contains more omega-3 fatty acids from fish, which lowers the risk of hypertension [15]. Some studies also showed that the carbohydrate-rich dietary pattern in the north is more likely associated with low HDL and higher risk of hypertension [13,33]. This overall dietary trend in China seems to correspond well with the recent economic development and increasing urbanization that has caused eating behavior change and increased oil consumption among the Chinese population [21].

Another important finding in our study is that the awareness of CDGs had no significant influence on lowering or increasing relative macronutrient intakes. The underlying reason for the lack of link between CDG knowledge and levels of macronutrient intakes is not yet understood, but it may be potentially explained by previous qualitative research on this issue showing that knowledge itself is not enough to influence food choice [26]. Some studies demonstrated that food choice can depend on the reliability of public information and a broader environment [34], suggesting that disseminating CDGs to the public alone would not significantly improve the nutrient status of Chinese population.

In terms of priorities of a healthy diet, adults who had a higher priority in a healthy diet were more likely to meet the DRI for macronutrients compared to those who had lower priority in a healthy diet. This could be interpreted by one study that demonstrated that personal evaluation or belief in foods, in our case, the priority given to healthy diet, can influence personal food choice [26]. Since one’s food attitude and values placed in a healthy diet can be developed over time [26], it is crucially important to implement dietary education and intervention targeting people of younger age. Moreover, the finding that females are having more fat intake than males suggests that women are at higher risk of developing obesity and other diet-related NCDs compared to men in China. The public health and nutrition policy might need to enhance the emphasis on maternal diet.

The strengths of this study include the use of a relatively large sample size and individual dietary data to assess the overall macronutrient intake status across two regions in China. It reveals the variations in the macronutrient intake status among Chinese adults with respect to different sociodemographic, socioeconomic, and lifestyle factors, providing insights into a more detailed and contextual understanding of adult nutritional status in different regions and areas. To the best of our knowledge, this is the first study that incorporated the evaluation of the link between an individual’s diet knowledge and belief system and macronutrient intake status among Chinese adults.

Nevertheless, our study contains several limitations. Firstly, the CNHS study did not update the most recent years of nutrition and health survey data since the year 2011; therefore, the study year 2011 used in this study is not likely to be the most representative of the current status of macronutrient intake among Chinese adults. Additionally, the 24 h dietary data collection method was largely dependent on recall, which is subject to inaccuracy. Secondly, the predictor variables selected in this study were somewhat subjective, including the selection of social and lifestyle factors most relevant to the study, and re-labeling and categorization of some variables. Moreover, the selection of the eight provinces that were divided into two regions in this study might not be generalized to the entire country. Third, the nutritional assessment in this study was mainly based on macronutrient composition that was documented to have different possible health outcomes [35], and thus led to bias without taking specific food groups into account. The analysis of the quality of the food intakes and dietary pattern in the further study would provide a better understanding of the nutritional status in China. Since physical activity also seems to be a significant factor influencing dietary intake, it is valid to expand knowledge in this area [36]. The inclusion of other important variables, such as anthropometric measurements and family incomes [37], in future studies might be more useful and informative for making public health interventions and nutrition policy. Overall, this study adds to the growing body of evidence linking macronutrient intake status with various important social and lifestyle factors.

## 5. Conclusions

Using the dietary data from the China Health and Nutrition 2011 Survey, we evaluated macronutrient intake status among Chinese adults (≥20 years) in association with different sociodemographic, socioeconomic, and lifestyle factors. More than half of the adults met the DRI for protein intakes, but more than half of the adults had high fat intakes and low carbohydrate intakes. Only fourteen percent of the adults simultaneously met the DRI for all three macronutrients. Relative carbohydrate intakes were inversely associated with relative fat intakes. Higher relative fat and lower relative carbohydrate intakes were more common among younger adults who lived in urban areas of the southern region, had higher socioeconomic level, and placed high priority in healthy diet. Having the knowledge of China Dietary Guidelines had no strong influence on macronutrient intakes. These findings suggest the inequality of macronutrient intake status among different age groups, genders, regions, and socioeconomic status. Our results indicate the need for further analysis of intake status of specific food groups and dietary patterns, as well as for development of region-specific dietary guidelines aiming to improve dietary quality of vulnerable populations.

## Figures and Tables

**Figure 1 nutrients-13-03044-f001:**
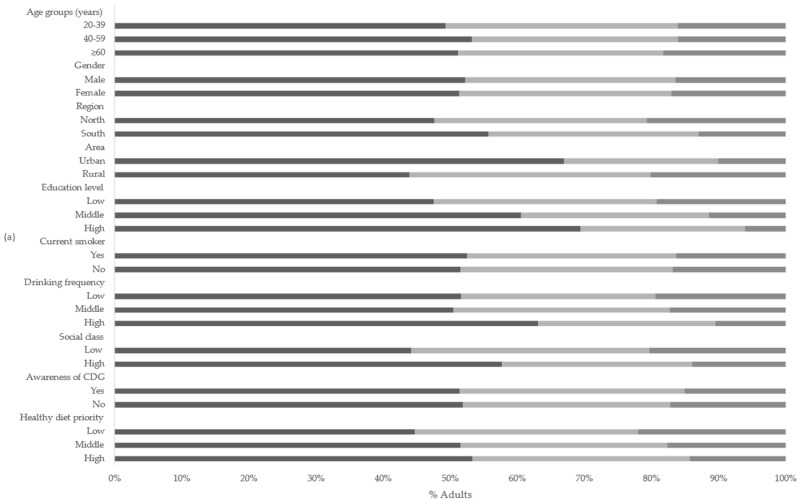
Percentage of adults with macronutrient intake below/meeting/above the recommended intake of DRI: (**a**) carbohydrates; (**b**) fats; (**c**) proteins.

**Figure 2 nutrients-13-03044-f002:**
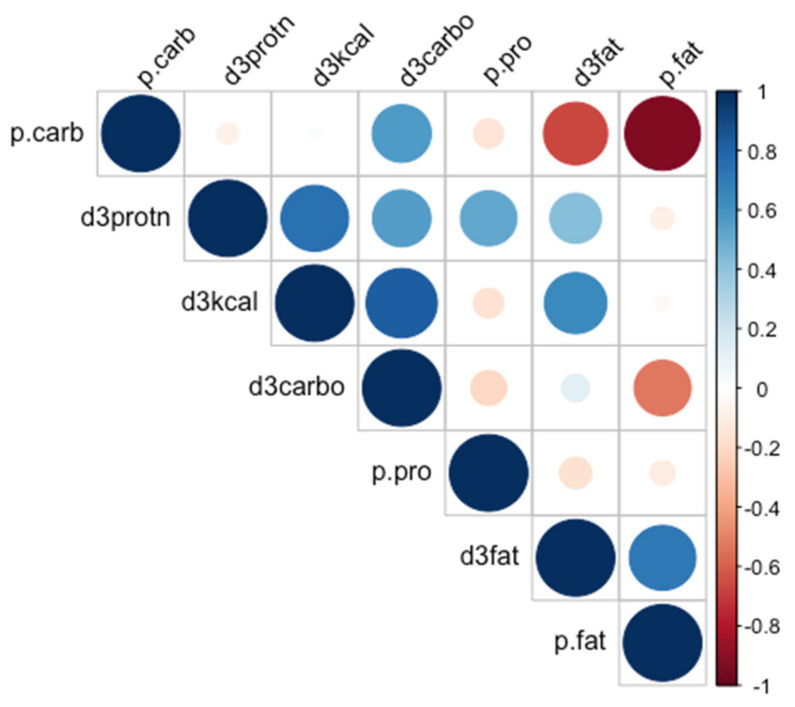
Correlations among total daily energy intake, macronutrient intakes, and relative macronutrient intakes. P.carb, percentage of daily energy from carbohydrate; p.pro, percentage of daily energy from protein; p.fat, percentage of daily energy from fat; d3carbo, three-day average intake of carbohydrate (g); d3fat, three-day average intake of fat (g); d3protn, three-day average intake of fat (g).

**Table 1 nutrients-13-03044-t001:** Total energy intake, carbohydrate, fat, and protein intakes of adults.

	N	%	Energy (kcal)	Carbohydrate (g)	Fat (g)	Protein (g)	Energy from Carbohydrate (%)	Energy from Fat (%)	Energy from Protein (%)
**Age groups (years)**									
20–39	1686	21.5	1996.7 (549.8)	268.5 (93.3)	71.1 (29.4)	65.1 (22.4)	53.7 (10.8)	32.2 (10.6)	13.2 (3.0)
40–59	3720	47.3	1991.8 (572)	272.9 (95.9)	69.1 (30.1)	67.2 (22.2)	54.7 (11.2)	31.3 (10.9)	13.6 (3.1)
≥60	2454	31.2	1825.4 (560.9)	246.8 (93.3)	65.3 (29.9)	58.3 (21.2)	54.04 (11.7)	32.3 (11.5)	13.0 (3.0)
**Gender**									
Male	3603	45.8	2101.0 (575.2)	283.5 (97.8)	73.0 (30.2)	68.5 (22.8)	53.92 (11.2)	31.4 (10.7)	13.2 (3.0)
Female	4257	54.2	1807.7 (528.1)	245.1 (89.2)	65.4 (29.3)	59.1 (20.8)	54.11 (11.3)	32.5 (11.3)	13.2 (3.1)
**Region**									
North	3821	48.6	1881.2 (566.7)	260.7 (92.3)	64.0 (30.2)	60.6 (21.4)	55.6 (11.2)	30.6 (1.21)	13.0 (2.9)
South	4039	51.4	1999.8 (565.7)	264.6 (97.7)	73.4 (29.1)	66.1 (22.7)	52.5 (11.1)	33.4 (10.7)	13.4 (3.1)
**Area**									
Urban	2673	34.0	1775.6 (548.1)	222.0 (82.3)	69.3 (31.0)	62.6 (23.8)	50.2 (11.2)	34.9 (11.1)	14.2 (3.3)
Rural	5187	66.0	2028.1 (560.8)	283.8 (94.5)	68.7 (29.5)	63.8 (21.4)	56.0 (10.8)	30.5 (10.7)	12.7 (2.8)
**Education level**									
Low	5761	73.3	1942.0 (575.1)	267.4 (97.2)	67.3 (29.9)	62.1 (21.7)	55.0 (11.4)	31.4 (11.2)	12.9 (2.9)
Middle	1455	18.5	1958.0 (554.9)	255.0 (88.6)	72.4 (30.3)	66.8 (23.1)	52.2 (10.6)	33.3 (10.5)	13.8 (3.2)
High	644	8.2	1908.4 (547.5)	238.1 (85.8)	74.3 (29.2)	67.6 (22.9)	49.8 (10.1)	35.2 (10.2)	14.3 (3.2)
**Current smoker**									
Yes	1994	25.4	2095.7 (585.7)	281.1 (98.0)	72.9 (72.9)	67.9 (22.7)	53.7 (11.5)	31.4 (10.8)	13.1 (3.0)
No	5866	74.6	1890 (554.0)	256.4 (93.3)	67.5 (29.6)	61.9 (21.8)	54.1 (11.2)	32.2 (11.1)	13.2 (3.1)
**Drinking frequency**									
Low	335	4.3	1871.5 (539.3)	254.3 (94.5)	66.0 (29.1)	64.6 (23.1)	54.2 (11.8)	31.8 (11.4)	14.0 (3.5)
Middle	6769	86.1	1920.3 (558.4)	261.9 (94.9)	68.4 (29.9)	62.7 (23.1)	54.4 (11.1)	32.1 (11.0)	13.2 (3.0)
High	756	9.6	2169.0 (625.1)	273.5 (97.2)	74.1 (30.6)	69.0 (23.5)	50.7 (11.8)	31.1 (11.0)	12.9 (3.2)
**Social class**									
Low	3464	44.1	2032.9 (580.3)	284.0 (97.8)	68.3 (29.9)	65.0 (23.5)	55.8 (10.9)	30.4 (10.7)	12.9 (2.8)
High	4396	55.9	1870.7 (549.9)	245.9 (89.6)	69.3 (30.1)	62.2 (22.9)	52.6 (11.3)	33.3 (11.1)	13.4 (3.2)
**Awareness of CDG**									
Yes	1858	23.6	2040.9 (583.8)	275.6 (95.5)	72.3 (30.5)	66.8 (23.0)	54.0 (10.7)	31.9 (10.4)	13.2 (3.1)
No	6002	76.4	1911.6 (561.2)	258.7 (94.7)	67.8 (29.8)	62.4 (21.9)	54.0 (11.4)	32.1 (11.2)	13.2 (3.0)
**Healthy diet priority**									
Low	514	6.5	1899.4 (576.4)	262.9 (94.2)	64.8 (31.8)	60.8 (21.4)	55.7 (12.4)	30.5 (12.1)	12.9 (3.1)
Middle	4535	57.7	1918.2 (571.2)	259.5 (95.7)	68.3 (30.3)	62.1 (22.3)	54.0 (11.5)	32.2 (11.3)	13.08 (3.0)
High	2811	35.8	1988.6 (561.9)	267.9 (94.2)	70.5 (29.0)	66.0 (22.1)	53.7 (10.6)	32.1 (10.4)	13.4 (3.0)

All values are means (standard deviation).

**Table 2 nutrients-13-03044-t002:** Demographic characteristics and chi-square tests of adults who simultaneously met the DRI for all three macronutrients.

	*n*	%	𝜒^2^	*p*
**N = 1609**				
**Age group**				
20–39	0	0.00	939.13	<0.001
40–59	266	24.90		
≥60	803	75.10		
**Gender**				
Male	481	45.00	10.71	<0.005
Female	1128	55.00		
**Region**				
North	541	50.60	0.16	0.69
South	1068	49.40		
**Area**				
Urban	220	20.60	370.10	<0.001
Rural	1389	79.40		
**Education**				
Low	825	77.20	942.86	<0.001
Middle	179	16.70		
High	605	6.10		
**Current smoker**				
Yes	796	24.00	290.22	<0.001
No	813	76.00		
**Drinking frequency**				
Low	85	8.00	1461.70	<0.001
Middle	945	88.40		
High	579	3.60		
**Social class**				
Low	1092	51.60	1.15	0.28
High	517	48.40		
**CDG knowledge**				
Yes	290	27.10	223.69	<0.001
No	1319	72.90		
**Healthy diet priority**				
Low	69	6.50	409.42	<0.001
Middle	605	56.60		
High	935	36.90		

**Table 3 nutrients-13-03044-t003:** Results of linear regression analysis of three macronutrient relative intakes and its associated factors (coefficients and 95% confident intervals).

	Univariate Simple Linear Model			Adjusted Multiple Linear Model		
	Coefficient	CI 95	*p*	Coefficient	CI 95	*p*
**% energy from carbohydrate**						
**Age groups (years)**						
≥60 (ref)						
40–59	−0.34	−0.91, 0.23	0.24			
20–39	0.65	0.05, 1.34	0.07			
**Gender**						
Male (ref)						
Female	0.18	−0.32, 0.68	0.47			
**Region**						
North (ref)						
South	−3.07	−3.56, −2.57	<0.001	−3.17	−3.67, −2.67	<0.001
**Area**						
Urban (ref)						
Rural	5.73	5.22, 6.24	<0.001	5.74	5.23, 6.25	<0.001
**Education level**						
Low (ref)						
Middle	−2.78	−3.42, −2.14	<0.001	−2.90	−3.55, −2.26	<0.001
High	−5.20	−6.11, −4.29	<0.001	−5.49	−6.41, −4.57	<0.001
**Current smoker**						
No (ref)						
Yes	−0.45	−1.02, 0.12	0.12			
**Drinking frequency**						
High (ref)						
Middle	3.66	2.81, 4.50	<0.001	3.69	2.81, 4.57	<0.001
Low	3.43	1.99, 4.87	<0.001	3.40	1.94, 4.87	<0.001
**Social class**						
High (ref)						
Low	3.20	2.70, 3.70	<0.001	3.55	3.03, 4.06	<0.001
**CDG knowledge**						
No (ref)						
Yes	0.02	−0.57, 0.61	0.95			
**Healthy diet priority**						
Low (ref)						
Middle	−1.69	−2.71, −0.66	<0.001	−1.70	−2.73, −0.68	<0.005
High	−1.98	−3.04, −0.92	<0.001	−2.02	−3.09, −0.96	<0.005
**% energy from fat**						
**Age groups (years)**						
≥40 (ref)						
20–39	−0.95	−1.54, −0.36	<0.005	−0.94	−1.54, −0.35	<0.005
**Gender**						
Male (ref)						
Female	1.12	0.63, 1.60	<0.001	1.11	0.62, 1.60	<0.001
**Region**						
North (ref)						
South	2.84	2.36, 3.33	<0.001	2.86	2.38, 3.35	<0.001
**Area**						
Urban (ref)						
Rural	−4.40	−4.91, −3.90	<0.001	−4.41	−4.91, −3.91	<0.001
**Education level**						
Low (ref)						
Middle	1.89	1.26, 2.52	<0.001	2.00	1.37, 2.64	<0.001
High	3.80	2.91, 4.70	<0.001	4.04	3.13, 4.95	<0.001
**Current smoker**						
No (ref)						
Yes	−0.80	−1.36, −0.24	<0.01	−0.80	−1.36, −0.24	<0.005
**Drinking frequency**						
High(ref)						
Low	1.04	0.21, 1.86	<0.05	1.04	0.22, 1.87	<0.05
**Social class**						
High (ref)						
Low	−2.99	−3.50, −2.50	<0.001	−3.00	−3.49, −2.52	<0.001
**CDG knowledge**						
No (ref)						
Yes	−0.17	−0.74, 0.41	0.57			
**Healthy diet priority**						
Low (ref)						
Middle	1.66	0.66, 2.67	<0.01	1.69	0.69, 2.70	<0.001
High	1.55	0.51, 2.58	<0.01	1.60	0.55, 2.64	<0.005
**% energy from protein**						
**Age groups (years)**						
≥60 (ref)						
40–59	0.31	0.16, 0.47	<0.001	0.32	0.17, 0.48	<0.001
20–39	0.72	0.52, 0.91	<0.001	0.72	0.53, 0.91	<0.001
**Gender**						
Male (ref)						
Female	0.04	−0.10, 0.17	0.61			
**Region**						
North (ref)						
South	0.36	0.23, 0.49	<0.001	0.37	0.23, 0.51	<0.001
**Area**						
Urban (ref)						
Rural	−1.50	−1.64, −1.36	<0.001	−1.50	−1.63, −1.35	<0.001
**Education level**						
Low (ref)						
Middle	0.90	0.73, 1.07	<0.001	0.94	0.75, 1.10	<0.001
High	1.39	1.15, 1.64	<0.001	1.46	1.20, 1.70	<0.001
**Current smoker**						
No (ref)						
Yes	−0.13	−0.29, 0.02	0.094			
**Drinking frequency**						
Low (ref)						
High	0.76	0.43, 1.10	<0.001	0.77	0.43, 1.10	<0.001
**Social class**						
High (ref)						
Low	−0.45	−0.58, −0.31	<0.001	−0.44	−0.58, −0.31	<0.001
**CDG knowledge**						
No (ref)						
Yes	0.06	−0.10, 0.22	0.44			
**Healthy diet priority**						
Low (ref)						
High	0.35	0.21, 0.49	<0.001	0.35	0.21, 0.49	<0.001

## Data Availability

The link to publicly archived datasets analyzed in this study is http://www.cpc.unc.edu/projects/china/about/proj_desc/survey (accessed on 28 July 2021).

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
