# Peer review of "Evaluation of Disparities in Adults’ Macronutrient Intake Status: Results from the China Health and Nutrition 2011 Survey"

_nutrients, 2021, doi:10.3390/nu13093044_

Round 1

Reviewer 1 Report

This retrospective study looks at patterns of macronutrient intake in certain specific regions in China. The work is well organized, and statistics seem solid. Yet the study presents observations without any truly novel outcomes, simply an overview of existing dietary patterns and adherence to dietary guidelines. While this may be useful as foundational information, it causes the article to be somewhat less important than other work dealing with more direct effects of nutrition and diet on health.

Some specific comments and questions:

MATERIALS AND METHODS

P2 L83: Missing data: Were missing data associated at all with demographics or other factors?

P2 L84: This is a result and does not belong here.

P3 L92: Household food composition: Where are these data? It is not clear how household food consumption was used in this study.

P3 L111: Final sample is also a result and does not belong in this section.

Figure 1: A dividing line or perhaps different color patterns for the provinces in N vs. S would be helpful and save the reader some time.

P4 L135: Were these cut points somehow used in the analysis? If so, how? It should be described here.

P4 L141: Why call this a ‘dummy’ variable?

P4 L166: Delete ‘whether’.

P4 L170: “For variables with more than two groups…” This is an awkward sentence. Please revise.

RESULTS

Table 1: This is a large table with lots of information. It may be easier to read if standard deviations were in parentheses and placed closer to their associated measure.

P6 L210: Referring to the next several paragraphs – can we assume that these findings were all significant? If so, that should be noted.

Figure 3: Labels need to be defined.

DISCUSSION

P14 L380: Not sure that this statement follows, especially since there did not seem to be an association between knowledge of guidelines and dietary patterns.

P14 L409: Should this perhaps be “developed over time”?

OVERALL

There is much discussion here of regional differences among northern and southern provinces (introduction, map), yet that factor (i.e. geography) becomes merely one cofactor among many and does not seem to be important, or at least is not explored as a significant contributor. It may be better to devote less time to the introduction of these regional effects, or alternatively, do a more thorough analysis of why there may be differences due specifically to regional effects. Other than that overall point, the remainder of the article is certainly well done.

Author Response

Dear reviewer,

Thank you so much for taking your precious time in reviewing our paper and providing valuable comments in such great details with such patience. It was your valuable and insightful comments that led to possible improvements in the current version. We have carefully considered the comments and tried our best to address every one of them. We hope the manuscript after careful revisions meet your high standards. 

Below we provide the point-by-point responses in red.

Point 1. This retrospective study looks at patterns of macronutrient intake in certain specific regions in China. The work is well organized, and statistics seem solid. Yet the study presents observations without any truly novel outcomes, simply an overview of existing dietary patterns and adherence to dietary guidelines. While this may be useful as foundational information, it causes the article to be somewhat less important than other work dealing with more direct effects of nutrition and diet on health.

Response 1. Great point. This article is simply an examination of the overall status of macronutrient intake among Chinese adults. Our original intention was indeed trying to make some analyses on specific health outcomes associated with the diet. However, we were not able to obtain specific anthropometric data from the CNHS 2011 survey year, since the year 2009 is the the only survey year with available anthropometric or biomarker data provided by the CNHS. Therefore, further study on nutrition and health among Chinese population in a more recent year will be done once the CHNS updates its biomarker dataset.

Point 2. (P2 L83) Missing data: Were missing data associated at all with demographics or other factors?

Response 2. Missing data or NA data were actually deleted during the data process, and they were no longer associated with these factors. We deleted the words "missing data" in the revised manuscript (P2 L88)

Point 3. (P2 L84) This is a result and does not belong here.

Response 3. Very good suggestion, but we intended to include this brief data summary in this section because we have read many other peer-reviewed articles that have done the very same thing in the material and method section. This result is simply a part of the data processing. However, we valued your point very much and decided to delete the result here in the revised manuscript (P2 90). 

Point 4. (P3 L92) Household food composition: Where are these data? It is not clear how household food consumption was used in this study.

Response 4. The household food data collection procedure was provided by the CNHS as shown in the reference No.19. After a careful consideration, we agreed to exclude this information because it was indeed not used in this study (revised manuscript P3 101)

Point 5. (P3 L111) Final sample is also a result and does not belong in this section.

Response 5. Good point. We deleted this information here in the revised manuscript (P3 135)

Point 6. Figure 1: A dividing line or perhaps different color patterns for the provinces in N vs. S would be helpful and save the reader some time.

Response 6. Fabulous idea. We modified this figure by adding the diving line at the beginning of revision, but we decided not to use figure 1 at all because we greatly value the overall comment you provided in the end regarding the regional effect. We decided to devote less time on introducing the regional effect and to treat it as one of the cofactors in this study.

Point 7. (P4 L135) Were these cut points somehow used in the analysis? If so, how? It should be described here.

Response 7. Good point. However, these cut points were to decide if the adults' macronutrient intakes are below/ meet/ above the DRI requirement. They were used in this analysis, and please refer to the results 3.4.

Point 8. (P4 L141) Why call this a ‘dummy’ variable?

Response 8. We called this a dummy variable because we used 1 to represent make and 2 to represent female. However, we decided to re-phrase it because it might be confusing (revised manuscript P3 132)

Point 9 P4 L166: Delete ‘whether’.

Response 9. Thank you a lot for this good call. Change has been made (P4 222)

Point 10 P4 L170: “For variables with more than two groups…” This is an awkward sentence. Please revise.

Response 10. Thank you a lot for pointing this out. Change has been made accordingly (P4 226)

Point 11Table 1: This is a large table with lots of information. It may be easier to read if standard deviations were in parentheses and placed closer to their associated measure.

Response 11. Thank you so much for this great suggestion. Change has been made accordingly (Figure 1)'

Point 12. P6 L210: Referring to the next several paragraphs – can we assume that these findings were all significant? If so, that should be noted.

Response 12. Thank you for giving this detailed comment. Results 3.1 to 3.3 were all simply summaries based on table 1. We added the reference to table 1 at several places in the revised manuscript (P5 L264; P6 L308; P6L320)

Point 13. Figure 3: Labels need to be defined.

Response 13. Thank you so much for this great suggestion. Change has been made accordingly (now it has been numbered as Figure 2 instead of figure 3)

Point 14. P14 L380: Not sure that this statement follows, especially since there did not seem to be an association between knowledge of guidelines and dietary patterns.

Response 14. Great point! The "education" here has more to do with education level in general, rather than the nutrition policy like CDG. We have re-phrased it (P14 L504).

Point 15. P14 L409: Should this perhaps be “developed over time”?

Response 15. Great call on this careless omission. Change has been made (P14 533)

Sincerely,

Yajie Zhao, Msc,

Department of Agricultural Sciences, Graduate School of Agricultural and Life Sciences, University of Tokyo

Reviewer 2 Report

General comments

The authors explored the cross-sectional association between macronutrient intake and sociodemographic characteristics such as region, area, education level, social class CDG knowledge, and healthy diet priority based on the large sample data in Chinese. The present study is the first report to compare the macronutrient intake between northern and southern Chinese regions. However, the scientific novelty from a global point of view was not sufficiently described.

Major comment 1

In the introduction section, although the author pointed out the association between macronutrient intake and non-communicable disease (lines 30-31), the importance of macronutrient intake is not fully described. The author should specifically explain how the over and under-intake of macronutrient increase the risk of each disease.

Major comment 2

The author explores the factors associated with each % energy from carbohydrates and % energy from fat. However, the relative proportion of carbohydrates, protein, and fat is not independent. So, the reviewer strongly recommends analysis investigating the factors associated with the proportion of people who simultaneously meet dietary reference intake of three macronutrients.

Minor comment 1

Figure 1 is too small to see.

Minor comment 2

Figure 2, the correlogram, needs a figure-legend to explain the abbreviated names of variables.

Minor comment 3

We can not determine the causal direction based on the cross-sectional study between macronutrient intake and CDGs. The term “effect” (in line 396) is not appropriate.

Author Response

Dear reviewer,

We appreciate your precious time in reviewing our paper and providing valuable comments. It was your valuable and insightful comments that led to possible improvements in the current version. We have carefully considered the comments and tried our best to address every one of them. We hope the manuscript after careful revisions meet your high standards. Below we provide the point-by-point responses that have been highlighted in red.

Point 1. General comments

The authors explored the cross-sectional association between macronutrient intake and sociodemographic characteristics such as region, area, education level, social class CDG knowledge, and healthy diet priority based on the large sample data in Chinese. The present study is the first report to compare the macronutrient intake between northern and southern Chinese regions. However, the scientific novelty from a global point of view was not sufficiently described.

Response 1. Great comment. This article is simply an examination of the overall status of macronutrient intake among Chinese adults. Our original intention was indeed trying to make some analyses on specific health outcomes associated with the diet. However, we were not able to obtain specific anthropometric data from the CNHS 2011 survey year, since the year 2009 is the the only survey year with available anthropometric or biomarker data provided by the CNHS. Therefore, further study on nutrition and health among Chinese population in a more recent year will be done once the CHNS updates its biomarker dataset.

Point 2. Major comment 1

In the introduction section, although the author pointed out the association between macronutrient intake and non-communicable disease (lines 30-31), the importance of macronutrient intake is not fully described. The author should specifically explain how the over and under-intake of macronutrient increase the risk of each disease.

Response 2. Thank you for this valuable suggestion. We have added a few specific literature review on this issue in the introduction part (Revised manuscript P1 L33-36

Point 3. Major comment 2

The author explores the factors associated with each % energy from carbohydrates and % energy from fat. However, the relative proportion of carbohydrates, protein, and fat is not independent. So, the reviewer strongly recommends analysis investigating the factors associated with the proportion of people who simultaneously meet dietary reference intake of three macronutrients.

Response 3. We strongly agreed. Great call. The further analysis on this matter has been made, and please refer to the newly-made table 2 on the revised manuscript (P5 L367-377).

Point 4. Minor comment 1

Figure 1 is too small to see.

Response 4. Good point. However, we modified this figure by enlarging it and adding the diving line at the beginning of revision, but we decided not to use figure 1 at all because we decided to devote less time on introducing the regional effect that seems to contribute only minor effect to the overall study and to treat it as one of the cofactors.

Point 5. Minor comment 2

Figure 2, the correlogram, needs a figure-legend to explain the abbreviated names of variables.

Response 5. Thank you so much for this great suggestion. Change has been made accordingly (now it has been numbered as Figure 2 instead of figure 3)

Point 6. Minor comment 3

We can not determine the causal direction based on the cross-sectional study between macronutrient intake and CDGs. The term “effect” (in line 396) is not appropriate.

Response 6. Great call on this word usage. Change has been made accordingly (P14 519)

Sincerely,

Yajie Zhao, Msc,

Department of Agricultural Sciences, Graduate School of Agricultural and Life Sciences, University of Tokyo

Round 2

Reviewer 2 Report

The author sufficiently responded to all comments, and the manuscript was revised appropriately.